# Engineered reversible inhibition of SpyCatcher reactivity enables rapid generation of bispecific antibodies

Christian Hentrich[1,4], Mateusz Putyrski[1,4], Hanh Hanuschka[1], Waldemar Preis[1], Sarah-Jane Kellmann[1], Melissa Wich[1], Manuel Cavada[1], Sarah Hanselka[1], Victor S. Lelyveld[2,3] & Francisco Ylera [1] ✉

The precise regulation of protein function is essential in biological systems and a key goal in chemical biology and protein engineering. Here, we describe a straightforward method to engineer functional control into the isopeptide bond-forming SpyTag/SpyCatcher protein ligation system. First, we perform a cysteine scan of the structured region of SpyCatcher. Except for two known reactive and catalytic residues, none of these mutations abolish reactivity. In a second screening step, we modify the cysteines with disulfide bond-forming small molecules. Here we identify 8 positions at which modifications strongly inhibit reactivity. This inhibition can be reversed by reducing agents. We call such a reversibly inhibitable SpyCatcher "SpyLock". Using "BiLockCatcher", a genetic fusion of wild-type SpyCatcher and SpyLock, and SpyTagged antibody fragments, we generate bispecific antibodies in a single, scalable format, facilitating the screening of a large number of antibody combinations. We demonstrate this approach by screening anti-PD-1/anti-PD-L1 bispecific antibodies using a cellular reporter assay.

SpyTag/SpyCatcher is a protein ligation technology in which a small peptide (SpyTag) and a small protein (SpyCatcher) react to form an isopeptide bond (Fig. 1A)[1]. Two improved versions of SpyCatcher have been engineered, both improving SpyTag-SpyCatcher affinity and accelerating the reaction velocity over the respective previous version[2,3]. SpyTag/SpyCatcher has proven extremely versatile, with applications for example in vaccine development, nanopore sequencing, or enzyme stabilization[4]. We have previously used this technology to develop a modular antibody platform, in which SpyTagged Fab antibody fragments can be rapidly combined with a wide variety of prefabricated SpyCatcher fusion proteins for site-specific labeling and/ or control of oligomeric state and antibody format[5].

Engineering functional control into proteins is an active field of research with implications for basic science and biotechnology[6]. Posttranslational modifications are a principal mechanism by which cells regulate protein function[7,8]. Such posttranslational modifications can naturally occur at cysteines, for example through reversible S-thiolation with glutathione[9]. Glutathionylation of the p50 subunit of NF-κB inhibits binding of this transcription factor to DNA; similarly, glutathionylation of MEKK1 kinase leads to its inactivation[10,11].

Cysteine scanning mutagenesis is an established method to interrogate protein function. It has been used for example to study the channel forming amino acids of ion pumps and channels, where solvent-exposed thiol groups react much faster with cysteine-reactive probes than buried residues[12]. In other studies, cysteines have been introduced close to the active site of enzymes, in order to conjugate libraries of small molecules for inhibitor screening, where local proximity would allow the identification of low-affinity compounds as a starting point for structure-activity relationship studies[13]. In one interesting example, cysteine scanning has been used to identify an

[1]Bio-Rad AbD Serotec GmbH, Anna-Sigmund-Str. 5, 82061 Neuried, Germany. [2]Department of Molecular Biology, Massachusetts General Hospital, Boston, MA, USA. [3]Department of Genetics, Harvard Medical School, Boston, MA, USA. [4]These authors contributed equally: Christian Hentrich, Mateusz Putyrski. ✉e-mail: francisco_ylera@bio-rad.com

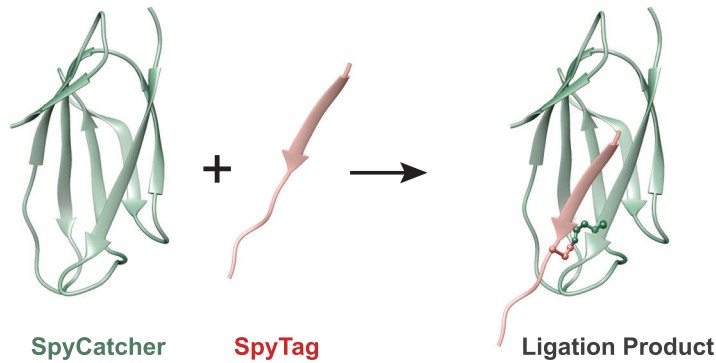

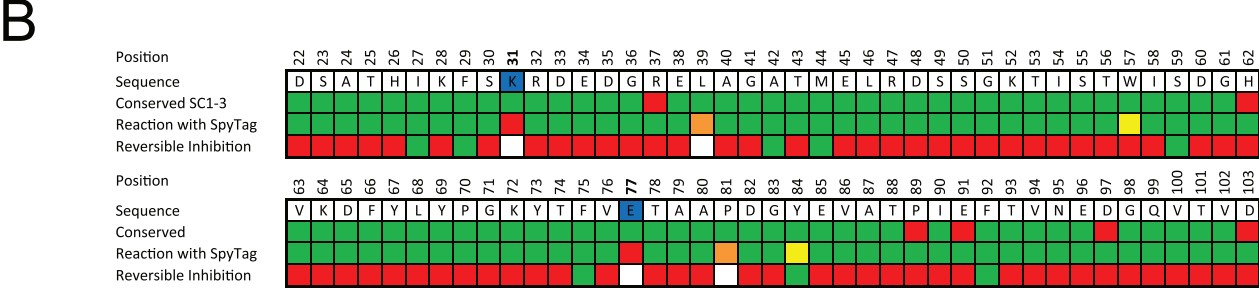

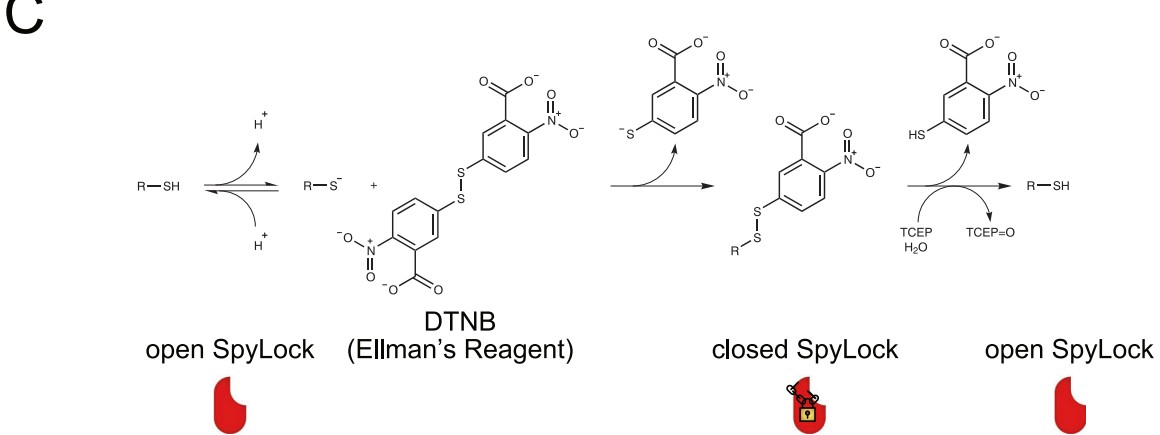

**Fig. 1 | Cysteine scan for reversibly inhibitable SpyCatcher003 mutants.**
**A** Model of the SpyTag-SpyCatcher reaction based on the crystal structure[20]. Spy-Catcher (green) reacts with SpyTag (red), forming a covalent isopeptide bond between aspartate 7 in SpyTag and lysine 31 in SpyCatcher. **B** Heat map depicting the cysteine scan of SpyCatcher003 for reversible inhibition upon modification with Ellman's reagent (DTNB). The first two rows show position and amino acid of SpyCatcher003, with reactive and catalytic residues in blue. The third row signifies amino acid conservation among SpyCatcher001, SpyCatcher002, and SpyCatcher003, using green for conserved and red for differing positions. The fourth row assesses reactivity of unmodified cysteine mutants with SpyTag within 90 min: green for fully reactive, yellow for slightly reduced reactivity, orange for strongly reduced reactivity, and red for unreactive. The fifth row evaluates reversible inhibition potential with Ellman's reagent: green for yes, red for no, and white for not tested. **C** Reaction of a cysteine residue with Ellman's reagent (DTNB) and reduction with TCEP.

allosteric site in a protease from dengue virus. 8 amino acids at suspected allosteric sites were mutated to cysteine. One of them, when modified with cysteine-reactive probes, quantitatively inhibited protease activity, and activity was restored after removal of the probe[14]. A similar approach was used to block a proton pump with a methanethiosulfonate, which could be removed by reduction[15].

Our purpose in this study is to engineer functional control into SpyCatcher. Specifically, we want to inhibit SpyCatcher in a reversible manner, so that we can activate SpyCatcher reactivity at will. Direct inhibition of SpyCatcher's active site has been achieved in a straightforward manner by protecting the isopeptide bond-forming lysine 31 with a photocleavable protection group through use of artificial amino acids[16,17]. We instead aim to regulate SpyCatcher activity indirectly through S-thiolation of engineered cysteines. Therefore, we perform a cysteine scan of every amino acid in the structured region of SpyCatcher to identify mutants with lowered activity when modified by disulfide-forming small probes. We identify several such positions and find the inhibition to be reversible upon reduction. We call such reversibly inhibitable SpyCatcher mutants "SpyLock".

To demonstrate the utility of this system, we use SpyLock to produce bispecific antibodies in a controlled stepwise manner. Bispecific antibodies are typically generated by fusing two distinct antibodies, yielding a single molecule capable of binding different epitopes, often on separate antigens. They have proven to be promising therapeutic agents, particularly in oncology[18]. Production of bispecific antibodies with correct light and heavy chain pairing is not trivial and has led to the development of a plethora of formats for bispecific antibody screening[19]. Utilizing a fusion of SpyLock and regular SpyCatcher, which we call "BiLockCatcher", we rapidly generate bispecific antibodies from SpyTagged Fab fragments and demonstrate their efficacy in cellular assays.

## Results

### Screening for reversibly inhibitable SpyCatchers

In order to engineer functional control into SpyCatcher, we focused on the ordered region of SpyCatcher according to the known crystal structure[20]. We chose SpyCatcher003, the fastest reacting SpyCatcher variant, as being able to control this version should have the highest interest for various applications. We therefore expressed 81 cysteine mutants of every amino acid in SpyCatcher003 between Asp22 and Asp103 in *E. coli*, except for S49C, which is an already described cysteine mutant used for site-specific labeling of SpyCatcher[21]. All cysteine mutants could be expressed and purified, with yields between 3.2 and 17.1 mg/L (median 6.9 mg/L) of purified protein from 50 mL expression cultures.

We then tested the ability of these mutated SpyCatchers to react with MBP (maltose binding protein) fused to SpyTag002. We used a SpyTag002 fusion due to a slightly higher bacterial expression yield compared to SpyTag003 fusion proteins[5]. Two SpyCatcher residues are essential for the isopeptide formation between the SpyTag and SpyCatcher: Lys31, which is the isopeptide bond-forming amino acid of SpyCatcher, and Glu77, which is a catalytic amino acid for the reaction[1]. As expected, cysteine mutations at these positions completely abolished the reactivity of SpyCatcher003. However, to our surprise, every other SpyCatcher003 mutant remained reactive, albeit in 4 cases (L39C, W57C, P81C, Y84C) with strongly reduced apparent reaction rates (Fig. 1B, Supplementary Fig. 1).

Next, we modified the cysteines of all SpyCatcher003 mutants with Ellman's reagent (DTNB, 5,5'-dithiobis(2-nitrobenzoic acid)), a disulfide bond-forming molecule that adds a relatively bulky and charged thionitrobenzoic acid group to the cysteine (Fig. 1C)[22]. We tested reactivity of such modified proteins and observed strongly reduced reactivity for 8 SpyCatcher mutants: I27C, F29C, A42C, M44C, S59C, F75C, Y84C, and F92C (Fig. 1B, Supplementary Fig. 2).

Effective control of SpyCatcher function requires this inhibition to be reversible. To test this, we took advantage of the reversible nature

of disulfide bond formation and incubated the 8 modified cysteine-SpyCatcher003 mutants with 10 mM TCEP (tris(2-carboxyethyl)phosphine), a common reducing agent. When removing the thionitrobenzoic acid group by reduction and restoring the free cysteine, reactivity with MBP-SpyTag002 was fully restored for all 8 mutants (Fig. 1B, Supplementary Fig. 3).

We thus have identified 8 different mutants of SpyCatcher003 that can be inactivated by reaction with Ellman's reagent and reactivated by subsequent reaction with a reducing agent and thereby established functional control of SpyCatcher's enzymatic activity. All 8 identified positions were conserved between SpyCatcher versions 1–3 (Fig. 1B).

In order to test whether this reversible inhibition was particular to Ellman's reagent or a more general phenomenon, we tested HPDP-biotin (N-[6-(biotinamido)hexyl]−3´-(2´-pyridyldithio)propionamide), which was designed to reversibly label cysteines. Indeed, despite not carrying a charge like TNB (5-thio-2-nitrobenzoic acid), HPDP-biotin was also able to reversibly inhibit the reactivity of the cysteine mutants (Supplementary Fig. 4). The first 5 mutations I27C, F29C, A42C, M44C, S59C were inhibited very strongly by both reagents, indicating that the loss of activity is independent of the charge of TNB but more likely due to the size of the introduced side chain. The last 3 mutations F75C, Y84C, F92C were more strongly inhibited by TNB, thus potentially dependent on charge for inhibition.

### Functional characterization of inhibitable SpyCatchers

When inspecting the positions of the 8 abovementioned mutations in the crystal structure (Fig. 2A), we noticed that 6 out of 8 wildtype residues were rather hydrophobic amino acids, and 5 of these pointed towards the hydrophobic core of the beta-barrel at the center of the SpyCatcher's Ig fold. Because of its missing SpyTag beta strand, the melting temperature of uncoupled SpyCatcher is much lower than after coupling with SpyTag[5]. We measured the melting temperature of the 8 identified cysteine mutants in unmodified form, after modification with HPDP-biotin, and post-reduction of the HPDP-biotin modified protein (Fig. 2B). For unmodified SpyCatcher mutants, all variants with cysteine replacing hydrophobic or aromatic amino acids had decreased melting temperatures, as one would expect from modifying amino acids directed towards the hydrophobic core (Fig. 2A). The A42C and S59C mutants were slightly stabilizing. Biotin-modified SpyCatcher mutants were always destabilized compared to the respective unmodified mutant. This was especially pronounced for the two stabilizing mutants A42C and S59C, with the Tm lowered by 9 °C and 14 °C, respectively. However, the fact that a clear melting temperature was measurable for each mutant demonstrates that both modified and unmodified SpyCatcher mutants remain folded at room temperature. The melting temperature returned to that of the unmodified protein once biotin was removed by reduction. Reaction speeds of all reversible inhibitable mutants were lowered compared to wildtype SpyCatcher003 (Supplementary Fig. 5, Supplementary Table 1) but remained sufficient for effective SpyTag coupling at low micromolar concentrations within one hour.

Among the 8 identified reversibly inhibitable SpyCatchers, we selected the S59C mutation for further study. This decision was based on several factors: the S59C mutation is the most conservative, given that serine and cysteine differ by only a single atom; it demonstrated robust reversible inhibition when exposed to Ellman's reagent or HPDP-biotin (Fig. 3A); the mutation is located outside the protein's hydrophobic core, minimizing aggregation potential; and it exhibited the highest stability of the mutants as indicated by its melting temperature.

When a reversibly inhibitable SpyCatcher mutant is in its disulfide-modified form, it is in a closed, locked state, preventing function. In contrast, a cysteine mutant of SpyCatcher in its reduced form is akin to an open, unlocked state, allowing it to react freely. Given this

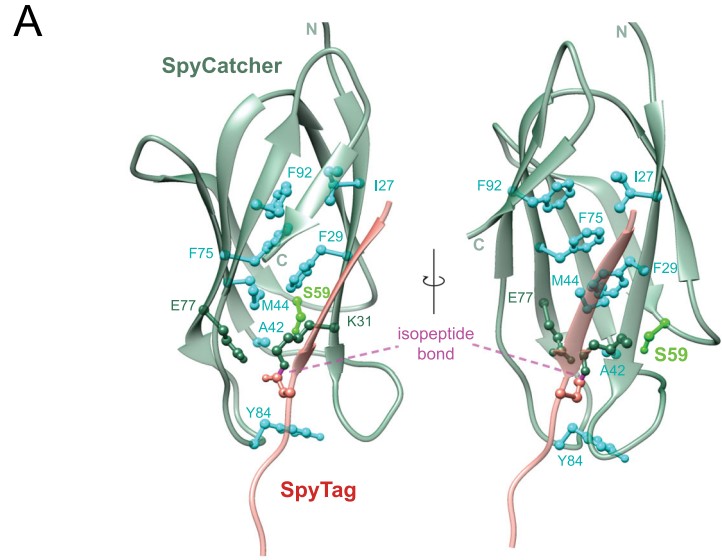

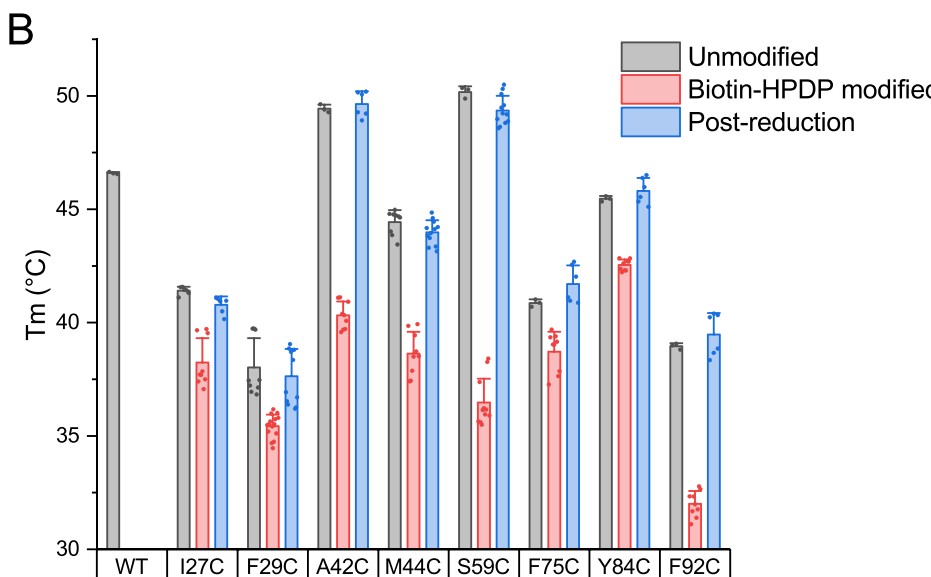

**Fig. 2 | SpyLock-enabling residues within the SpyCatcher structure and the thermal stability of corresponding cysteine mutants. A** Model of SpyCatcher001 (green) and SpyTag001 (red) with the residues capable of reversible inhibition upon substitution with cysteine highlighted in green (serine 59) or cyan. The catalytic residue is dark green, the isopeptide bond is labeled. Two views, rotated by 90 degrees. **B** Melting temperatures (determined by nanoDSF) of all identified reversibly inhibitable cysteine mutants in their unmodified and HPDP-biotin-blocked form (before and after reduction), in comparison to wildtype Spy-Catcher003. Error bars indicate standard deviations, while the bars represent the mean of the individual measurements, which are shown as points (3–15 replicates per condition). Source data are provided as a Source Data file.

functional resemblance to a lock, we have named reversibly inhibitable SpyCatcher mutants 'SpyLock'. In this manuscript, we usually refer to the S59C mutant of SpyCatcher003 as SpyLock and use the terms "open SpyLock" for the reactive, reduced version with a free thiol group and "closed SpyLock" for the unreactive disulfide version.

We analyzed the time course of coupling reactions of closed SpyLock (treated with HPDP-biotin) in presence and absence of TCEP (Fig. 3B). After one hour, in absence of TCEP, almost no reaction occurred, whereas the reaction was close to completion in presence of the reducing agent. Upon overnight incubation, a small portion of the closed SpyLock reacted also in the absence of TCEP, indicating that rather than fully inhibiting the reaction, the biotinylation of cysteine 59 in SpyLock slows down the reaction by orders of magnitude. As expected from the fact that all 8 amino acid positions responsible for SpyLock behavior are conserved between SpyCatcher001, 002, and

003, S59C mutants of SpyCatcher001 and 002 work equivalently (Fig. 3C).

Since proteins with free cysteines tend to dimerize by forming intermolecular disulfide bonds in non-reducing environments, we tested the dimerization potential of SpyLock by incubating open SpyLock with or without MBP-SpyTag002 in PBS/0.5 mM NiCl$_2$ for 24 h. As a control we used SpyCatcher003 with an N-terminal cysteine (Cys-SpyCatcher003). In the absence of MBP-SpyTag002, both SpyLock and Cys-SpyCatcher003 dimerized almost quantitatively. However, there was a strong difference in dimerization status of SpyTag002-coupled Cys-SpyCatcher003 and SpyLock. Cys-SpyCatcher003-SpyTag002 again dimerized almost quantitatively, whereas there was only a very small amount of SpyLock-SpyTag002 dimerization (Fig. 3D). As expected, in contrast to Cys-SpyCatcher003, pre-dimerized SpyLock could not react with SpyTag002 anymore.

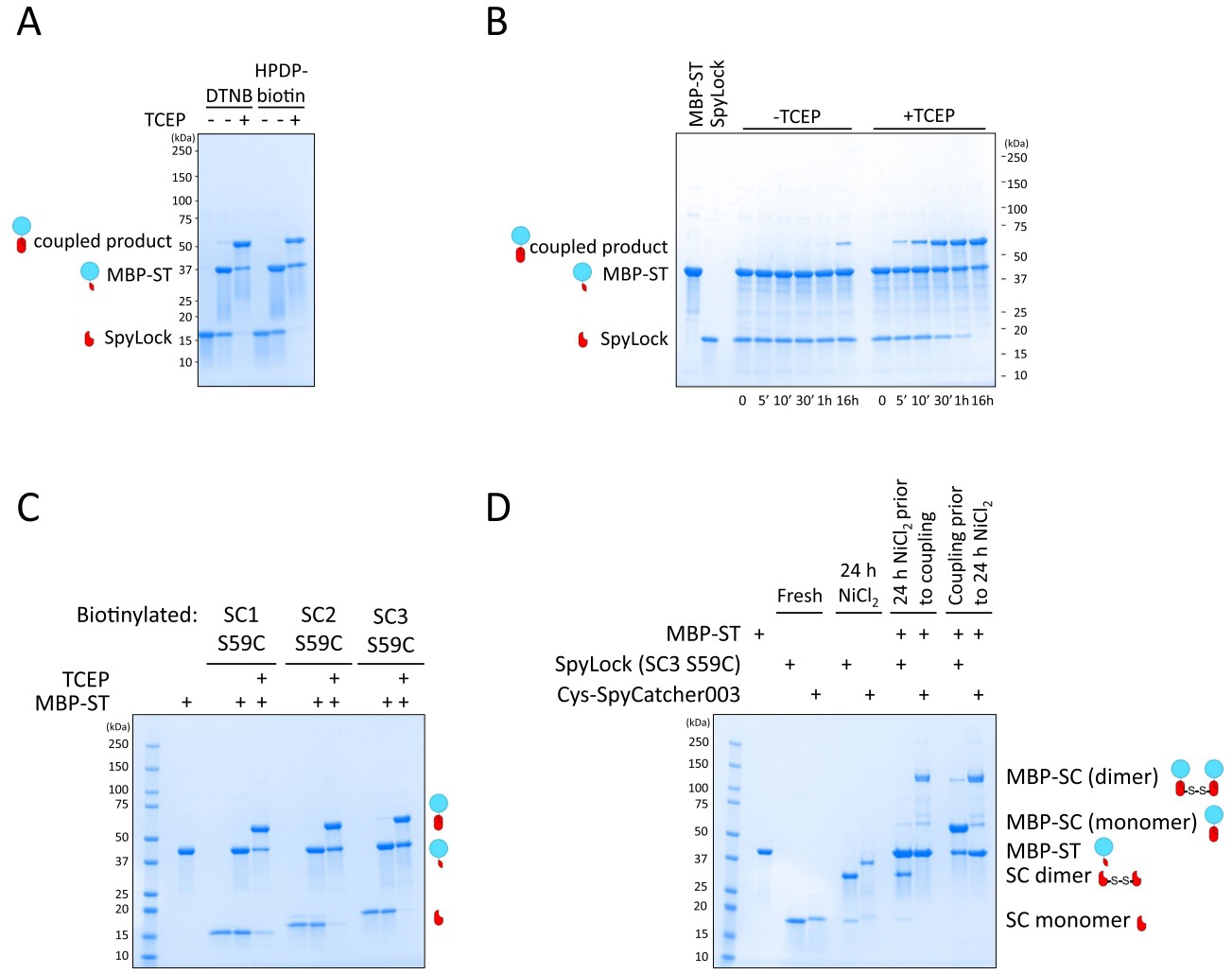

**Fig. 3 | SpyLock coupling reactions. A** Reducing SDS-PAGE gel showing the result of reacting 4 μM SpyCatcher003 S59C (SpyLock) modified with either DTNB or HPDP-biotin with 6 μM MBP-SpyTag002 (MBP-ST) in presence or absence of the reducing agent TCEP (5 mM) for 90 min. **B** Time course of the reaction of 4 μM biotin-SpyCatcher003 S59C (biotin-SpyLock) with 6 μM MBP-SpyTag002 (MBP-ST) in presence or absence of the reducing agent TCEP (5 mM), visualized on a reducing SDS-PAGE gel. **C** Reducing SDS-PAGE gel of the S59C variants of SpyCatcher001/2/3 (SC1-3, 4 μM) modified with HPDP-biotin, reacting with 6 μM MBP-SpyTag002

(MBP-ST) for 90 min in presence or absence of the reducing agent TCEP (5 mM). **D** Non-reducing SDS-PAGE analysis of NiCl₂-induced dimerization of SpyLock or N-terminal Cys-SpyCatcher003 and reaction of dimerized catchers with MBP-SpyTag002. Dimerization occurred over 24 h at 37 °C. Reaction conditions were identical for pre- and post-dimerization coupling: 10 μM SpyCatcher, 15 μM MBP-SpyTag002, 90 min. Each experiment shown in **A–D** was performed at least twice independently, a representative gel is shown.

## Molecular dynamics simulations

To investigate the underlying mechanism of the SpyLock inhibition we performed molecular dynamics (MD) simulations of all 8 SpyLock mutants. Beginning from a high-resolution crystal structure model of SpyCatcher, we generated derivative models for each of the 8 cysteine mutants and their corresponding biotinylated form. Perturbations in the overall tertiary fold were minimal over short 10 ns trajectories for most of these mutants, but the conformation of the disulfide-bridged biotin modification showed significant flexibility owing to the long 12-atom linker in HPDP-biotin. For 6 out of 8 modified mutants, the conformation of the modeled biotin modification generally occluded the SpyTag binding pocket (Fig. 4A, B, D, F−H). This observation is wholly consistent with a simple steric mechanism for SpyCatcher inhibition in the modified form of these 6 cysteine mutants.

In contrast, disulfide modification of the two remaining mutants, A42C (Fig. 4C) or S59C (Fig. 4E), caused more subtle changes on the reactive residues in the SpyTag binding pocket in MD simulations. For the loop mutant A42C, biotin modification at this position distorted the adjacent strand E, which in turn allosterically impacts the

conformation of the critical catalytic residue Glu77 (Fig. 4C, red arrow). Ser59 stands out as the most distal mutant site relative to the SpyTag binding pocket and catalytic residues. Interestingly, the effects of modification at S59C on the binding pocket are minimal in the simulated model. Rather than disrupting the binding pocket directly, the biotin modification extends through a solvent accessible pore formed at this loop (amino acids 59−63) from the protein surface distal to the binding pocket (Supplementary Fig. 6A, B).

Experimentally, we examined the interaction between SpyLock S59C and SpyTag002 using biolayer interferometry (BLI, Supplementary Fig. 7). In these experiments we observe a combination of reversible binding and non-reversible isopeptide bond formation. During the association phase, SpyCatcher003 led to a strong and rapid signal increase, which was less intense and slower when open SpyLock S59C was used. This is consistent with the lowered reaction rate for the S59C mutant observed in bulk. The closed SpyLock S59C, modified with either biotin or TNB modifications, shows almost no binding to the sensor. Initially, this finding appears contradictory to our computational models, which predicted an accessible SpyTag binding groove.

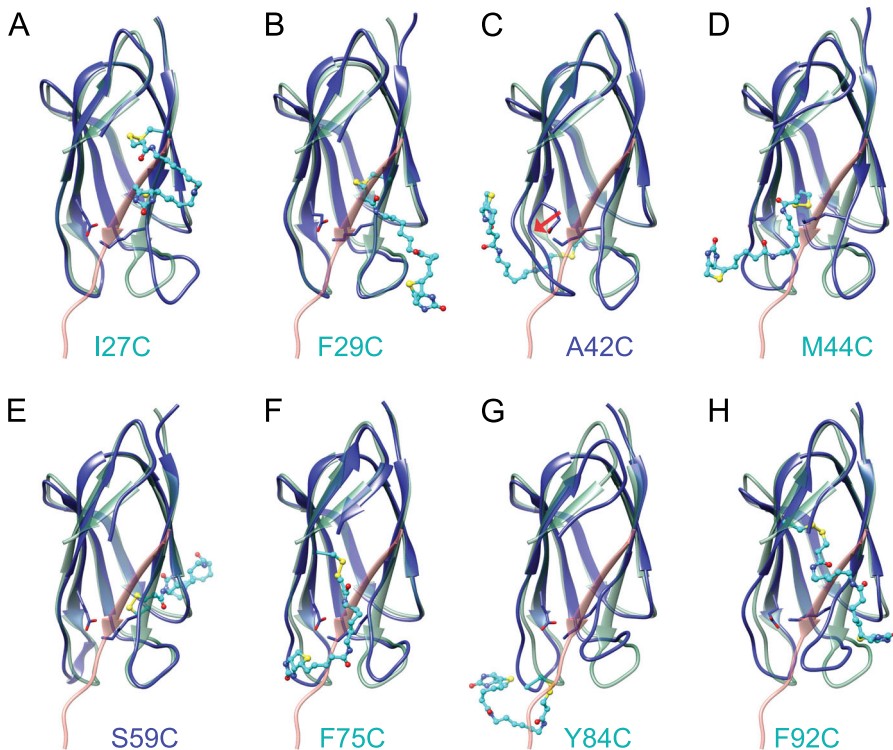

**Fig. 4 | Molecular modeling of modified SpyCatcher mutants. A–H** Snapshots of the indicated mutant models following 10 ns simulations at 300 K, displayed as blue ribbon cartoons with side chain conformations shown only for the modified residue (cyan) and key catalytic residues Glu77 and Lys31 (blue). For each mutant, the SpyCatcher-SpyTag structure (PDB 4MLI) is superimposed in green with the SpyTag strand highlighted (red). Models with steric clashes between the biotin modification and SpyTag are labeled in cyan, models without steric clashes in blue. Red Arrow in **C** points to SpyCatcher strand E where the catalytic Glu77 is displaced.

To address this discrepancy, we conducted extended 100 ns simulations of SpyCatcher003 wildtype, S59C, and S59C-biotin in the presence of bound SpyTag at elevated temperature (Supplementary Fig. 6C–E). In simulations at 350 K, presence of the modification was associated with greater instability in the SpyTag conformation at the reaction center. In particular, the Cα distance between the reactive SpyCatcher lysine (Lys31) and SpyTag aspartate (Asp7) was substantially destabilized over the simulation trajectory in the presence of the biotinylation vs. the S59C mutation alone (Supplementary Fig. 6F). Although these simulations are unlikely to reach equilibrium on a computationally accessible timescale, these results at short times are consistent with a reduced SpyTag binding capability of the biotin-modified S59C mutant.

**Deactivation of SpyLock reactivity within an assay**

To explore whether SpyLock reactivity could be turned off directly during an assay, we performed ELISAs with directed immobilization via SpyLock. We coated a fast-reacting SpyLock (M44C) on a Maxisorp plate, followed by titration of a SpyTagged protein, SpyTag002-sfGFP (Supplementary Fig. 8). When using SpyTagged antibodies for detection, the ELISA failed due to reaction of the antibody with the open SpyLock on the surface. To test the ability of immobilized unreacted SpyLock to be switched off after sfGFP immobilization, the plate was incubated with 10 mM Ellman's reagent for 1 h. Indeed, this was sufficient to turn off SpyLock reactivity quantitatively, equivalent to reacting all open SpyLocks with excess SpyTag peptide, and allowed the ELISA to be performed successfully. This demonstrates that indeed SpyLock reactivity can be dynamically turned off within an experiment.

**SpyLock enables bispecific antibody generation**

To demonstrate SpyLock's potential, we employed it in the generation of bispecific antibodies. Therefore, we constructed a fusion protein of SpyCatcher and SpyLock, analogous to BiCatcher, which we call "BiLockCatcher"[5]. After locking with a disulfide-forming reagent, BiLockCatcher has two independent reaction sites for SpyTag, one of which is constitutively active, and one which is inactive and can be unlocked on demand. When a first SpyTagged antibody fragment is added, it will only react with the wild-type SpyCatcher moiety. Then, the SpyLock is unlocked by addition of 5 mM TCEP and a second SpyTagged antibody fragment is added, which can react with the open SpyLock moiety (Fig. 5A, B).

Using TCEP in the presence of antibodies raises the concern that the intradomain disulfide bonds of the antibodies are also reduced, and the antibodies thereby lose their function. The inter-chain disulfide between heavy and light chains is not required for the Fab stability and is not present in our SpyTagged Fab fragments[5]. To quantify the impact of TCEP, we incubated three Fab fragments with 5 mM TCEP for 16 h, the longest time Fabs would be exposed to TCEP during the generation of bispecific antibodies. This did not change their affinity as measured by BLI (Supplementary Fig. 9, Supplementary Table 2). Even incubating Fab fragments with 5 mM TCEP for an entire week had mostly minor effects on antigen binding ability as measured by titration ELISA (Supplementary Fig. 10). Nevertheless, even though the amount of TCEP present in the samples would therefore not be problematic for most applications, we still decided to routinely remove it by quenching with the non-toxic bis-PEG$_3$-azide, as described[23]. This procedure lowers the amount of measurable free TCEP by 97.3%, as determined by use of Ellman's reagent.

While the abovementioned protocol for assembly of bispecific antibodies is very fast, in practice it is sometimes challenging to achieve very high purity of the bispecific antibodies. This issue arises from the difficulty in determining protein concentrations with sufficient accuracy to ensure precise 1:1 Fab:SpyCatcher coupling. To address scenarios where high purities are essential, we developed a purification protocol that takes advantage of the possibility of using HPDP-biotin with SpyLock (Fig. 5C). In the first step, we coupled biotin-

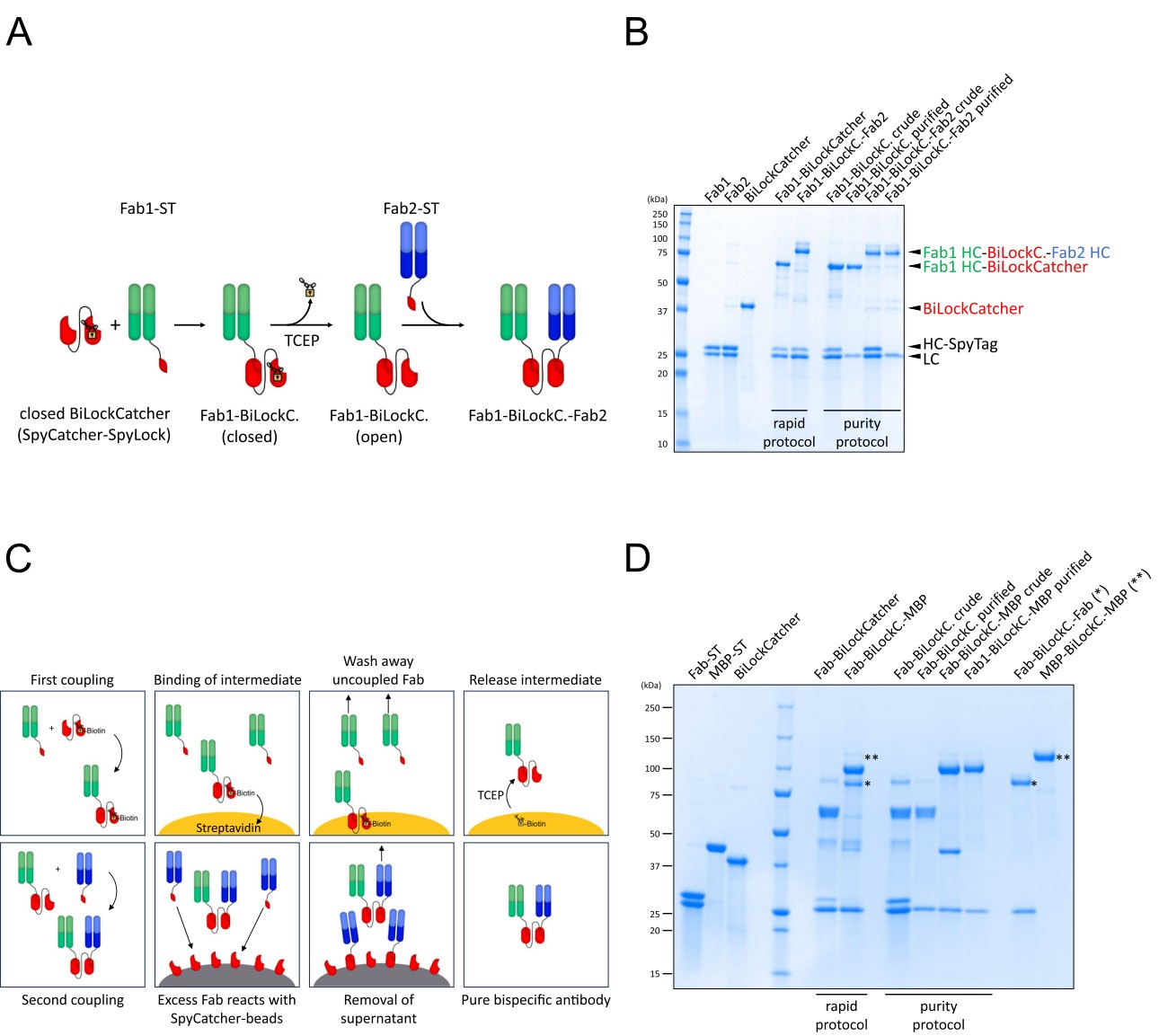

**Fig. 5 | Bispecific antibody generation with BiLockCatcher. A** Scheme of the reaction of BiLockCatcher (SpyCatcher-SpyLock, red) with two different Fabs (green, blue) to create a bispecific antibody. The disulfide-forming small molecule is signified by a lock. ST: SpyTag. **B** Reducing SDS-PAGE gel showing sequential generation of bispecific antibodies by either a rapid protocol or a slower protocol involving two purification steps. Rapid protocol: 3 μM BiLockCatcher, 3 μM Fab1, 3 μM Fab2, first coupling 45 min. Purity protocol: 3 μM BiLockCatcher, 4.5 μM Fab1, 4.5 μM Fab2. First coupling 60 min. Second coupling was carried out overnight in both cases. HC heavy chain, LC light chain. This experiment was carried out 3 times, representative gel is shown. **C** Scheme explaining the purity protocol involving two sequential purification steps. Order left to right, top to bottom. **D** SDS-PAGE to assess purity of bispecific antibodies generated with the rapid or purity protocol.

Gel analogous to **B**, but instead of two Fabs, one SpyTagged Fab (Fab-ST) and MBP-SpyTag002 (MBP-ST) were used, allowing to visualize bivalent monospecific contaminants at different apparent molecular weight than the bispecific product. BiLockCatcher based on SpyCatcher002 was used to avoid the formation of double bands (Supplementary Fig. 14). Monospecific standards Fab-BiLockCatcher-Fab and MBP-BiLockCatcher-MBP are loaded in the last two lanes. Corresponding contaminations are indicated by asterisks. Rapid protocol: 10 μM BiLockCatcher, 10 μM Fab, 10 μM MBP, first coupling 45 min. Purity protocol: 10 μM BiLockCatcher, 15 μM Fab, 15 μM MBP, first coupling 150 min. Second coupling was carried out overnight in both cases. This experiment was performed 3 times, representative gel is shown.

BiLockCatcher with an excess of the first antibody (usually 1.5 molar equivalents) to ensure complete coupling of the SpyCatcher003 within one to three hours. Then, we incubated the product with streptactin or streptavidin beads to pull down the biotinylated BiLockCatcher-Fab#1 and washed away excess Fab#1. Any present non-biotinylated BiLockCatcher that might have coupled twice to the same antibody is also removed in this step. We then incubated the solid phase with TCEP, thereby simultaneously releasing BiLockCatcher-Fab#1 from the solid phase and unlocking the reactivity of the SpyLock moiety. Next, BiLockCatcher-Fab#1 was incubated with Fab#2, also in 1.5-fold molar excess, usually overnight. Afterwards, the bispecific antibodies with excess Fab#2 were incubated with beads on which

SpyCatcher003 was covalently immobilized. Unreacted excess Fab#2 reacted with the SpyCatcher003 beads and, after TCEP quenching, the supernatant contained highly pure bispecific antibodies. We measured the yield of this purification protocol using SpyTag002-sfGFP and fluorescence measurements of samples on SDS PAGE gels, in order to quantify only the coupling products and not any protein contaminants that are purified away during the procedure. The first step has a yield of 72%, the second step a yield of 80% (Supplementary Fig. 11).

The similarity in size between two Fabs makes it impossible to unambiguously demonstrate the purity of bispecific products using gel electrophoresis. To address this, we employed SpyTagged coupling partners of distinct sizes, a Fab and MBP, to create a bispecific

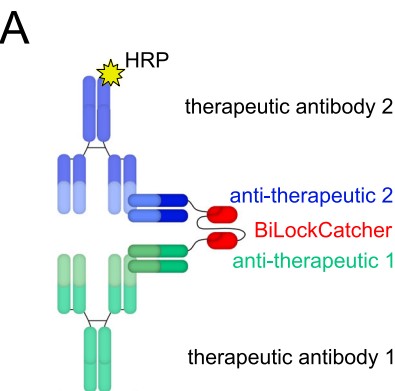

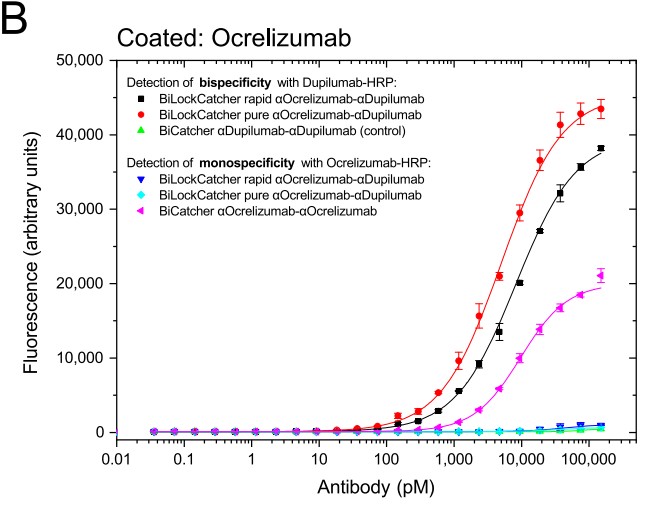

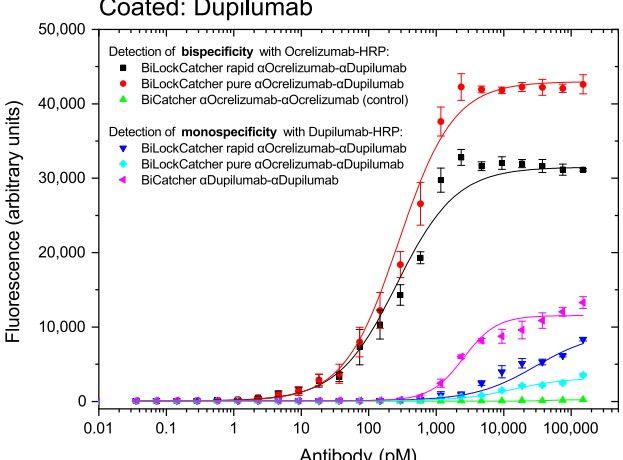

**Fig. 6 | Bispecificity demonstrated by bridging sandwich ELISA.** Bispecific anti-ocrelizumab/anti-dupilumab antibodies were produced according to the rapid or purity protocol and their serial dilutions were tested in bridging sandwich ELISA in which antigens dupilumab or ocrelizumab were coated on the plate and HRP-labeled form of either of both antigens was used for detection. **A** Schematic representation of the assay. **B** ELISA assay to detect bispecific antibodies. Left: ocrelizumab coated. Right: dupilumab coated. Monospecific bivalent controls constructed with BiCatcher3 were also tested with both HRP-labeled antigens. Coating antigen: 1 μg/mL, sandwich titration in 1:2 dilution series starting from 150 nM, HRP-labeled detection antigen: 2 μg/mL. Measurements performed independently in triplicates with the exception of non-cognate background controls, for which only one measurement was performed. Logistic fits were applied; error bars represent standard deviations, points represent the mean of measurements. Source data are provided as a Source Data file.

MBP-Fab fusion. This allowed for the straightforward differentiation between bivalent monospecific and bispecific products. While there is contamination of the monospecific products visible with the fast protocol, they are essentially absent from the purified samples (Fig. 5D).

**Functional validation of bispecific antibodies generated with BiLockCatcher**

To validate the functionality of our bispecific antibodies, we performed bridging ELISAs with ocrelizumab and dupilumab as antigens, with one antigen coated on plate and the second HRP-labeled antigen in solution. Our bispecific antibodies, created from two anti-idiotype antibodies, bridge them successfully, while the monospecific bivalent controls do not (Fig. 6).

Programmed cell death protein 1 (PD-1) and its ligand PD-L1 are common targets for cancer immunotherapy and interact with a low intrinsic affinity of around 8 μM[24]. Bispecific targeting of PD-1 and PD-L1 has shown enhanced anti-tumor activity in mouse models when compared to combined antibody treatment or individual monotherapies[25]. To ensure compatibility of BiLockCatcher-based bispecifics with cellular assays, we generated bispecific antibodies from known therapeutic anti-PD-1 and anti-PD-L1 antibodies and performed a sandwich staining on cells: PD-1-overexpressing HKB11 cells were incubated with bivalent monospecific or bispecific antibodies, then with PD-L1-biotin, and finally were stained with streptavidin-PE followed by flow cytometry. All bispecific constructs stained the cells, whereas none of the bivalent monospecific controls did (Supplementary Fig. 12).

To assess the utility of bispecific SpyLock-based antibodies in cellular screening with more complex assays, we generated 36 bispecific antibodies from 4 anti-PD-1 and 9 anti-PD-L1 antibodies originating from the Pioneer phage display library (Bio-Rad). We used a PD-1/PD-L1 immune checkpoint inhibitor screening assay based on reporter Jurkat-Lucia TCR-hPD-1 cells and Raji-APC-hPD-L1 antigen presenting cells (Invivogen). In this assay, a blockade of the PD-1/PD-L1 interaction results in luciferase expression (Fig. 7A). All 36 bispecific antibodies exhibited a dose-dependent response, with luciferase signal commencing at a concentration of 10 nM. Notably, some constructs had already reached their maximal signal level at this concentration (Fig. 7B).

We characterized the assay performance of one anti-PD-1/anti-PD-L1 bispecific pair in greater detail (Fig. 7C). We created the bispecific antibody in both possible reaction orders, i.e., coupling either the anti-PD-1 or the anti-PD-L1 antibody first. As controls, we included a monospecific bivalent anti-GFP antibody created with BiLockCatcher,

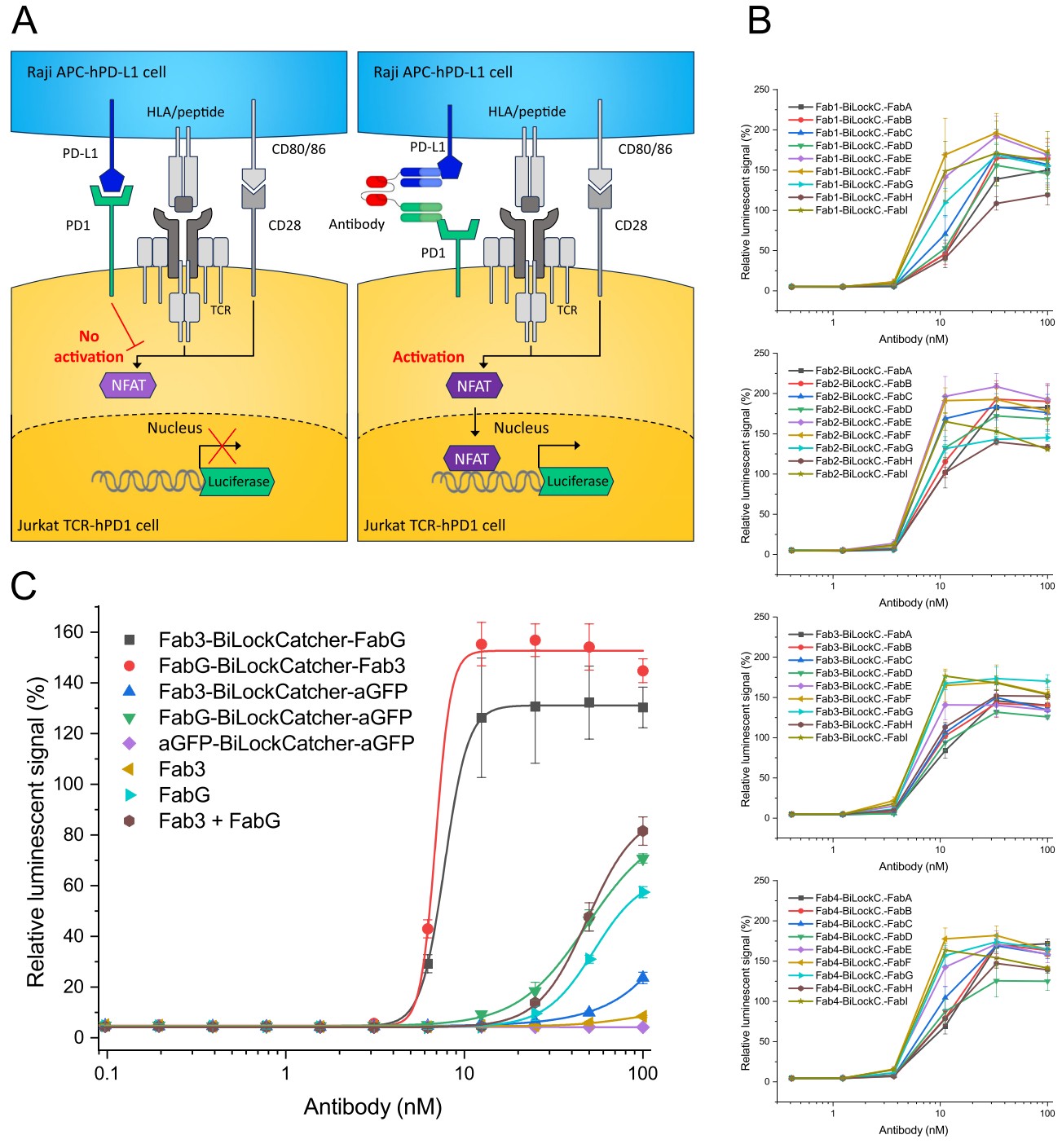

**Fig. 7 | Bispecific antibodies in cellular bioluminescent PD-1/PD-L1 blockade assay. A** PD-1/PD-L1 assay scheme. Left: unperturbed PD-1/PD-L1 interaction inhibits TCR signaling. Right: blocking of the PD-1/PD-L1 interaction with bispecific antibodies leads to transcription of the reporter gene. **B** 4 anti-PD-1 antibodies (Fab 1 – 4) and 9 anti-PD-L1 antibodies (Fab A - I) derived from the Pioneer library were used to construct 36 bispecific antibodies (purity protocol) and tested in PD-1/PD-L1 blockade assay. Results are shown in separate graphs corresponding to 4 anti-PD-1 antibodies tested. Source data are provided as a Source Data file. **C** PD-1/PD-L1 blockade assay performance of one bispecific antibody in comparison to individual Fabs, an equimolar mixture of Fabs (each at the indicated concentration), and additional controls with an irrelevant antibody (anti-GFP). For the anti-PD-1/anti-PD-L1 bispecific antibody, both coupling orders were tested. Logistic fit was applied. All measurements (in **B** and **C**) were conducted in triplicate and are presented as the mean ± SD, expressed as a percentage of the signal generated by treating the cells with 50 nM dostarlimab. Source data are provided as a Source Data file.

anti-GFP/anti-PD-1 or anti-GFP/anti-PD-L1 bispecific constructs, as well as monovalent Fabs. In monovalent Fab format, the anti-PD-1 antibody led to a very low response, while the anti-PD-L1 antibody elicited a modest response at high concentrations. The mixture of the two Fabs gave a roughly additive response compared to the individual Fabs. However, in bispecific format, the assay response was very robust, with

signal plateau already reached at 10 nM as in the initial screening experiment. The bivalent BiLockCatcher-based anti-GFP monospecific control did not yield a detectable signal. However, the bispecific anti-PD-1/anti-GFP or anti-PD-L1/anti-GFP gave slightly higher responses than their monovalent equivalents. This could be due to the fact that the larger constructs are better able to sterically block the PD-1/PD-L1

interaction on the assay cells. In summary, the data show a strong inhibitory effect of the bispecific antibodies compared to their corresponding Fabs.

For practical applications, the ability to produce and store large batches of closed BiLockCatcher over extended periods would be advantageous. To explore this possibility, we conducted accelerated stability studies on BiLockCatcher closed with either Ellman's reagent or HPDP-biotin. The BiLockCatcher modified with Ellman's reagent demonstrated instability, reverting from its closed state within just two weeks of incubation at 20 °C. In contrast, modification with HPDP-biotin maintained a stable closed state for over 16 weeks at the same temperature (Supplementary Fig. 13).

## Discussion

We have employed an easy and straightforward strategy to engineer functional control into SpyCatcher. It consisted of conducting a cysteine scan, i.e., producing a cysteine mutant for each amino acid, and testing for functionality before and after modification with a disulfide-forming reagent. We expect this approach to be adaptable to other split proteins like SnoopCatcher, DogCatcher, or split GFP 1-10/11[26–28]. These proteins are likely very suitable for the engineering-in of reversible inhibition by cysteine scanning, since usually inaccessible amino acids are exposed to solvent in split proteins and can readily react with disulfide-forming reagents.

Besides the known catalytic residue and the isopeptide bond-forming residue, we did not find a single cysteine mutation that abolished enzymatic function of SpyCatcher. However, we did uncover 8 positions that did severely inhibit activity upon modification with a disulfide bond-forming small molecule.

Our MD simulations of SpyLocks modified with HPDP-biotin allow grouping mutants according to the likely mechanism of inhibition. I27C, F29C, M44C, F75C, Y84C, and F92C likely function by steric inhibition of SpyTag binding. In A42C and S59C, the biotin modification instead takes on a configuration that does not directly block the binding site. Rather, the modification leads to more subtle conformational changes in the simulations, suggesting that these SpyLocks might be considered as being under allosteric regulation. Notably, the melting temperature of SpyLocks A42C and S59C are lowered the most upon modification with HPDP-biotin, suggesting that a non-steric mechanism is more destabilizing – especially for S59C where our model shows that the biotin modification is threaded through a solvent-accessible pore in SpyCatcher.

It is possible that the precise nature of the inhibition depends not only on the position of the modified amino acid, but also on the nature of the modification. For example, we observed differences in the efficiency of inhibition with Ellman's reagent and HPDP-biotin at some positions (Supplementary Fig. 4). On the other hand, every position found in the screen with Ellman's reagent also showed some level of inhibition with HPDP-biotin, suggesting a certain generality of the SpyLock positions.

Reasoning that more conservative mutations should be beneficial with regards to long-term stability, we decided to focus on the serine 59 to cysteine SpyLock. With just one atom of difference, the reaction speed of this variant is considerably slower compared to that of the wild-type SpyCatcher. This could be due to weakened hydrogen bonding interactions formed by the mutant side chain at this position, leading to subtle conformational shifts in the SpyCatcher reaction center[29,30]. In the structure of SpyCatcher, Ser59 is oriented inward, and our MD simulations strongly suggest that this positioning is preserved in its cysteine mutant form, SpyLock S59C. This inward orientation conceals the sulfhydryl group after coupling with SpyTag, accounting for the slow rate of dimer formation compared to proteins with more exposed cysteine residues. This property is highly desirable in SpyLock to keep it in the intended monomeric state after reduction.

Efforts to control SpyCatcher reactivity have been developed in the past. One straightforward alternative way to inhibit SpyCatcher reactivity is to modify the reactive lysine of SpyCatcher. This has been demonstrated particularly with photocleavable protection groups[16,17]. While this allows very elegant applications such as the generation of 3D structures in hydrogels, the SpyLock approach is much easier to implement and more economical to scale as it does not require an expression system capable of incorporating artificial amino acids. It can be produced in bacteria and reacted with inexpensive small molecules in bulk. Light-controlled reactivity in general is extremely powerful in specific research settings, but it is not well adaptable to most routine laboratory workflows.

Other strategies using disulfides and redox potential for controlling the reactivity of SpyCatcher have been published, however, these approaches differ fundamentally from our work[31,32]. Matsunaga et al. established a system based on split-Spy0128, which can be considered a predecessor of the SpyTag-SpyCatcher system[32]. Their approach relies on holding a non-reactive SpyTag-like peptide in the SpyCatcher-like binding pocket with a disulfide bond, thus shielding the access to the reaction center. Upon reduction of the disulfide, the shielding peptide can be displaced by wildtype SpyTag-equivalent peptide and the reaction can occur. Wu et al. have developed a way to change the conformation of SpyTag in such a way that it is strained and cannot react anymore[31]. Upon cleavage of a disulfide bond, the SpyTag can relax and react with SpyCatcher. In yet another example, SpyTag incorporated into a light-sensitive protein remains unreactive in the dark and becomes reactive after unfolding of the protein through irradiation with blue light[33]. All these approaches are highly engineered specialized applications, whereas our SpyLock approach relies only on a single engineered cysteine and is probably generalizable to other split proteins. Another interesting approach based on a non-reactive SpyTag has recently been developed by the Howarth laboratory[34]. Here, instead of using a disulfide, a non-reactive SpyTag is held in place by a protease-cleavable linker. Upon cleavage by a protease, the non-reactive SpyTag dissociates and unmasks SpyCatcher reactivity.

A notable limitation of SpyLock is its reliance on purified proteins, rendering it incompatible with in vivo use. Another constraint for SpyLock technology is the use of thiolation and reducing agents. SpyLock fusion proteins containing reduced cysteines will be thiolated during the protection step. This will be reversed during the deprotection step but still might be a concern in some cases. For BiLockCatcher, the first coupled protein needs to be able to withstand a low concentration of reducing agent for one hour, before it is quenched. We have shown that this is unproblematic for the internal disulfide bonds of Fabs, but it may be a concern for other proteins. In this study, we have purified each cysteine mutant for the cysteine scan. It may however also be possible to perform an in vitro selection combined with next generation sequencing to identify reversibly inhibitable split proteins, similar to techniques used for binding interface mapping[35].

In this work, we used Ellman's reagent and HPDP-biotin to generate the disulfide required for inhibiting the reactivity of cysteine mutants of SpyCatcher. Depending on the position of the cysteine, we could see differences in locking efficiency. More noticeable was the stability of the disulfide bond over time. Whereas BiLockCatcher blocked with HPDP-biotin remained stable, BiLockCatcher-TNB was unstable over time and therefore not suitable for bulk preparations. We expect that many disulfide-forming small molecules and also peptides would be suitable to inhibit SpyLock reactivity. Besides biotin, activated variants of other small molecules compatible with their cognate affinity purification reagents such as maltose, azide, alkines, or glutathione can be envisioned. Also activated thiol-reactive peptide tags (e.g. 3-nitro-2-pyridinesulfenyl-activated Strep-tag) could be an option. However, broad availability and low cost are good arguments for the molecules we have chosen in this study.

A straightforward application of SpyLock is the generation of bispecific antibodies. The therapeutic potential and efficacy of bispecific antibodies is widely acknowledged[18]. The best antibody pairs for bispecific antibodies are commonly identified experimentally and the screening space grows quadratically with the number of antibody sequences to be tested in combination - in one study, 40,000 bispecific antibodies were examined[36]. As most final therapeutic bispecific antibody formats are labor intensive to produce, it is useful to reduce the number of bispecific antibodies that need to be converted into the final format. Therefore, a first screening step in an alternative format to validate general functionality can massively reduce the workload and accelerate candidate selection. To this end, many different approaches have been developed, using protein ligation technologies such as split inteins, sortase A, transglutaminase, bispecific Fab-SpyCatcher-Fc fusion proteins, or SpyCatcher-SnoopCatcher fusions[19].

Our SpyLock strategy has several key differences from most of these approaches. The biggest advantage of using a SpyLock-SpyCatcher dimer is the ability to use only a single antibody format –only SpyTagged antibodies are required, and they can be produced rapidly from bacteria[5]. This advantage is especially pronounced when the antibodies are directly generated with a SpyTag, such as we have established for Pioneer and HuCAL antibody libraries[37,38], but it is easy to reclone any human antibody fragment for SpyTagged expression in *E. coli*. Alternatively, SpyTagged antibody fragments can be expressed in eukaryotic cell culture. Another strength of our approach is the possibility to perform the bispecific antibody generation in two ways. Our rapid approach only involves the sequential mixing of few reagents and can be performed within 90 min.

Alternatively, due to the presence of biotin on the closed SpyLock, we are able to couple with excess of antibody, drive the reaction to completion and purify away any unreacted antibody monomers as well as antibody dimers. The second coupling can also be carried out with excess of the second antibody, and leftover second antibody can easily be removed due to its reactive SpyTag. This procedure strongly increases the purity of the bispecific antibody, and, while more complex, can also be scaled up and automated easily. An optional buffer exchange step at the end could be used to remove any remnant TCEP or TCEP-quencher. This purification protocol allows a more efficient use of monomeric Fabs compared to Driscoll et al., which requires a 5-fold Fab excess in the second coupling step[34]. The amount of Fab required becomes an important economical factor when producing a large number of bispecific antibody pairs. However, the introduction of a biotinylated cysteine or Avi tag[39] in the protease sensitive linker applied by Driscoll et al. would allow the use of our two-step purification protocol.

SpyLock for generation of bispecific antibodies is open to implementation in many different geometries. As demonstrated by Driscoll et al., the precise geometry of the catcher-based bispecific antibodies can have a profound impact on the functionality of the reagents[34]. This again exemplifies the necessity of final verification of the selected antibody pair in the format in which the bispecific antibody will be therapeutically administered. We chose here a construct analogous to BiCatcher, with a flexible linker sequence[5]. This linker is readily adjustable in terms of length and stiffness, enabling the exploration of various geometries and mimicking the typical paratope reach distance of different antibody formats. To further expand the set of SpyLock-based bispecific formats for screening, it should also be possible to combine SpyLock with knob-in-hole Fc domains, with the caveat that exposed disulfides in the hinge region should either be avoided or, alternatively, reoxidized after bispecific generation[40]. Less canonical formats, such as SpyCatcher-Fc-SpyLock for tetravalent bispecific antibodies should also be easy to implement. SpyLock significantly expands the toolbox for the modular construction of antibodies based on SpyTag/SpyCatcher, providing a straightforward pathway to bispecific antibodies. We expect that the introduction of redox control of protein ligation through engineered reversible inhibition will prove useful in other fields.

## Methods

### Construct design and cloning

DNA sequences of the cysteine mutants of SpyCatcher and the SpyLock-SpyCatcher fusion constructs were ordered from Twist Bioscience either as gene strands, followed by Gibson assembly into pET-28a(+), or as genes cloned into pET-28a(+). SpyCatcher mutant positions are shown in Fig. 1B.

SpyCatcher sequences have been described previously[1–3]. The BiLockCatcher sequence is identical to previously described BiCatcher sequence, with an S59C mutation in the second SpyCatcher[5]. For the coupling purity assay, BiLockCatcher based on SpyCatcher002 was used to avoid the appearance of double bands that occur with SpyCatcher003 (Supplementary Fig. 14). Plasmids were transformed into BL21 cells for expression.

### Fab and SpyCatcher purification

Expression of Fabs, SpyTagged Proteins, and SpyCatchers was performed as described[5]. Briefly, *E. coli* strains SK13 carrying pBBx2 plasmids encoding Fabs, or BL21 carrying pET28a plasmids encoding SpyCatchers/SpyLocks, were grown overnight at 27.5 °C or 30 °C, respectively, in 250 mL cultures. Cells were harvested by centrifugation, frozen, and lysed with BugBuster (Merck KGaA), supplemented with 2 mg/mL lysozyme (Roche). Purification was performed using Ni-NTA agarose (Qiagen) as the matrix, with 20 mM sodium phosphate, pH 7.4, 500 mM sodium chloride, and 30 mM (wash) or 250 mM (elution) imidazole as buffers. Finally, proteins were rebuffered into PBS using PD10 columns (Cytiva). BiLockCatcher mean yield per liter of expression culture was 23 mg/L (SD: 9 mg/mL).

All Fab antibodies were validated in a quality control ELISA, in which they were tested for specific binding of cognate recombinant antigens together with at least three irrelevant antigens as negative controls.

### SpyCatcher003 beads

SpyCatcher003 was immobilized on Profinity Epoxide Resin (Bio-Rad) according to the manufacturer's instructions. In short: 1.9 g of dry resin was weighed out and allowed to swell in an excess of coupling buffer (50 mM boric acid, 0.5 M $K_2SO_4$, pH 9.0 with NaOH). 97 mg of SpyCatcher003 at 6 mg/mL in coupling buffer was added to 12 mL of settled resin and mixed for 22 h at room temperature. The coupled resin was washed with PBS and remaining active groups were blocked by addition of 40 mL of 1 M ethanolamine, pH 9.0 and 5 h incubation at room temperature. After washing with PBS and 100 mM sodium phosphate, 500 mM NaCl, pH 6.0, and subsequently with 100 mM sodium phosphate, 500 mM NaCl, pH 8.0, the resin with 7 mg/mL immobilized SpyCatcher003 was stored in PBS at 4 °C.

### Screening for reversible inhibition

In the initial screening phase, we incubated SpyCatcher cysteine mutants at a concentration of 4 µM and MBP-SpyTag002 at a concentration of 6 µM in PBS at room temperature for one hour to evaluate their reactivity. Samples were then boiled in SDS sample buffer, followed by SDS-PAGE. In the second screening step, SpyCatchers were incubated with 2.5 mM TCEP for 1 h to fully reduce the cysteines. This was followed by buffer exchange with PD MiniTrap G-25 columns (Cytiva) into 100 mM sodium phosphate buffer, pH 8.0. Resulting protein concentrations were determined by A280 measurements. Subsequently, all SpyCatchers were incubated for 1.5 h at room temperature with 430 µM DTNB (Thermo Fischer, from a 10 mM stock in 100 mM sodium phosphate, pH 8.0), equivalent to a tenfold molar excess of the highest concentrated SpyCatcher sample. To assess reactivity of the modified SpyCatchers, 4 µM of TNB-reacted

SpyCatchers were incubated with 6 µM MBP-SpyTag002 for 1.5 h at room temperature, followed by SDS-PAGE. In the third screening step, mutant SpyCatchers that showed strong inhibition when modified with TNB were tested for reversibility of the inhibition. 4 µM TNB-SpyCatchers were incubated for 1.5 h with 6 µM MBP-SpyTag002 in presence or absence of 10 mM TCEP (sufficient to reduce any excess DTNB) and analyzed by SDS-PAGE.

### Generation of closed SpyLocks

SpyLock constructs (including BiLockCatcher fusions) were incubated in PBS/2.5 mM TCEP for one hour, followed by buffer-exchange (MiniTrap G-25 columns, Cytiva) into DTNB-labeling buffer (100 mM sodium phosphate, 1 mM EDTA, pH 8.0) or biotin-labeling buffer (PBS pH 7.4, 1 mM EDTA). Fresh stock solutions of labeling reagents were prepared, 10 mM DTNB in 100 mM sodium phosphate, pH 8.0 or 4 mM EZ-Link HPDP-biotin (Thermo Fisher) in DMSO. A tenfold excess of labeling reagent was added to the SpyLock constructs and the labeling reaction was typically carried out overnight at room temperature. The next day, the buffer was exchanged to PBS and the SpyLock concentration was measured using a Bradford assay (Bio-Rad), using BSA as calibration standard. Modified SpyLocks were aliquoted and stored at −80 °C for long-term storage, or −20 °C for short-term storage.

### TCEP quantification

A standard curve was generated using known TCEP concentrations. The standard and samples were reacted with Ellman's reagent (Thermo Fisher) and photometrically measured, both in accordance with the manufacturer's instructions. TCEP concentrations in the samples were then calculated using linear regression based on this standard curve.

### Reaction rate and yield measurements

To maintain sfGFP fluorescence, reactions of SpyCatchers with SpyTag002-sfGFP were stopped by mixing with SDS sample buffer and incubation at 50 °C for 5 min, as described[3]. Fluorescence images of unstained SDS gels were recorded on a ChemiDoc MP (Bio-Rad) and band intensity was analyzed using ImageLab 6.1 (Bio-Rad).

### Antibody stability measurements by BLI or ELISA

Antibodies were incubated for 16 h at room temperature or one week at 4 °C, in presence or absence of 5 mM TCEP. Multi-cycle kinetic measurements were performed with BLI as described, using biotinylated mGFP as antigen[38]. Briefly, BLI measurements were performed on an Octet HTX instrument (Sartorius). Streptavidin sensors (SA, Sartorius) were loaded with biotinylated mGFP and quenched with 10 µg/mL biocytin in PBS. The baseline was set in running buffer (PBS/0.1% BSA/0.02% Tween 20). Three GFP-binding Fab fragments (AbD18705, AbD50087 and AbD51293), preincubated for 16 h in PBS with or without 5 mM TCEP, were measured at concentrations ranging between 0.3 and 80 nM. Association of the Fabs fragments was assessed for 10 min, followed by dissociation for 20 min. Reference measurement was performed in running buffer without any added Fab and was used for background subtraction for the Fab-containing samples.

For ELISAs, 2 µg/mL mGFP were coated on Maxisorp plates and ELISAs were performed as described[5]. Briefly, mGFP was coated on 384 well plates (Nunc Maxisorp MTP, Thermo Fisher) at a concentration of 1 µg/mL in PBS, 20 µL/well, at 4 °C overnight. After washing the plates 3x with PBST (PBS with 0.05% Tween 20 (Merck Millipore)), blocking with 5% BSA (Sigma-Aldrich) in PBST for 1.5 h at room temperature and washing again 3x with PBST, 20 µL/well of a 1:2 dilution series of the anti-GFP Fabs, starting at a concentration of 32 µg/mL, were transferred onto the plate. After 1 h of incubation at room temperature, the plates were washed five times with PBST before the addition of 20 µL/well of 2 µg/mL anti-Fab-HRP (STAR126P, Bio-Rad) secondary antibody, prepared in HISPEC assay diluent (Bio-Rad). After an incubation for 1 h at room temperature, plates were washed ten times with PBST, followed by detection with QuantaBlu fluorescence detection reagent (Thermo Fisher).

### Inactivation of SpyLock during ELISA

10 µg/mL SpyLock S44C in PBS/2.5 mM TCEP was coated overnight in Maxisorp plates (Thermo Fisher). After washing with PBS/0.05% Tween-20 (PBST), a 1:2 dilution series of SpyTag002-sfGFP in PBST, starting at 100 µg/mL, was incubated for 1 h. After washing with PBST, the plate was incubated for one hour with 100 mM sodium phosphate buffer, pH 8.0, supplemented with 10 mM Ellman's reagent or 1 µM SpyTag003 peptide depending on experimental condition. After washing with PBST, plates were incubated with 8 µg/mL of anti-GFP Fab-SpyTag002 (AbD50098ad) in PBST for 1 h. This was followed by washing, incubation with 2 µg/mL anti-Fab-HRP (STAR126P, Bio-Rad) in HISPEC buffer (Bio-Rad) for 1 h, washing with PBST, and detection with QuantaBlu detection reagent (Thermo Fisher).

### SpyLock-SpyTag BLI measurements

For determination of SpyLock-SpyTag interaction, BLI measurements were performed on an Octet RED384 instrument (Sartorius). Spy-Tag002 peptide with an N-terminal biotin (Intavis) was immobilized on streptavidin sensors (SA, Sartorius) in PBS with immobilization levels of 2 ± 0.5 nm. Quenching was carried out with 10 µg/mL biocytin in PBS. After setting the baseline in PBS/0.1% BSA/0.02% Tween 20/500 mM NaCl, association of SpyLock, SpyLock-TNB, SpyCatcher003 (100 µg/mL in PBS/0.1% BSA/0.02% Tween 20/500 mM NaCl) was assessed for 1 h, followed by dissociation in PBS/0.1% BSA/0.02% Tween 20/500 mM NaCl for 30 min. For reference, sensors were coated with biocytin only. Measurements with Biotin-SpyLock were carried out as above, except that biocytin was present in the assay buffer to prevent unspecific binding of the SpyLock to streptavidin sensors. Data were analyzed with Octet Analysis Studio software 12.2.

### Rapid bispecific antibody assembly

For the rapid bispecific antibody assembly an equimolar amount of closed BiLockCatcher and the first Fab-SpyTag002, both in PBS, were incubated for 30 min at room temperature in PBS. Then, 5 mM TCEP and an equimolar (regarding BiLockCatcher) amount of the second Fab-SpyTag002 were added and allowed to react for at least one hour at room temperature. This was followed by addition of 50 mM bis-PEG$_3$-azide (Lumiprobe, from undiluted stock) for TCEP quenching if so desired.

### Bispecific antibody assembly with purification

In this approach, the initial coupling of closed biotin-BiLockCatcher and the first Fab-SpyTag was carried out with 1.5-fold molar Fab excess for 1–3 h. Constructs were then added to pre-washed (PBST) and pre-blocked (5% BSA in PBST) magnetic streptavidin beads (Dynabeads M-280, Thermo Fisher) or Streptactin-agarose (IBA Lifesciences), with the amount of beads used being tenfold greater than what would be theoretically required based on the manufacturer's listed binding capacity. Beads were incubated for 1 h on a rotator, washed 3 times with PBST and incubated with 5 mM TCEP in PBS to release the SpyLock from the beads and at the same time open it. For larger batches of bispecific antibodies, since the first coupling is not yet in quadratic space, this is feasible even for large numbers of antibodies. With calculated concentrations based on the observation that the biotin pull-down and disulfide reduction are quantitative, the supernatant was split into equal parts for the second Fab coupling. The second antibody was added in 1.5× molar excess of the intermediate construct and coupled for at least 3 h, typically overnight. This was followed by addition of 50 mM bis-PEG$_3$-azide for TCEP quenching. To remove excess Fab 2, the bispecific antibodies were incubated with pre-washed SpyCatcher003 beads for 15 min and the purified bispecific antibody was collected in the flow-through.

## Sandwich ELISA

Capture antigens (ocrelizumab, dupilumab) were coated on 384 well plates (Nunc Maxisorp MTP, Thermo Fisher) at a concentration of 1 μg/mL in PBS, 20 μL/well. The next day, the plates were washed 3× with PBST (PBS with 0.05% Tween 20 (Merck Millipore)) and then blocked with 100 μL of 5% BSA (Sigma-Aldrich) in PBST for 1.5 h at room temperature and washed again 3× with PBST. Next, 20 μL of a 1:2 dilution series of the bispecific antibodies, starting at a concentration of 15 μg/mL, were transferred onto the plate and allowed to incubate for 1 h at room temperature. After incubation, the plates were washed five times with PBST before the addition of HRP-conjugated detection antibodies (ocrelizumab-HRP and dupilumab-HRP) with 20 μL/well and concentration of 2 μg/mL, prepared in HISPEC assay diluent (Bio-Rad). HRP conjugates were prepared using the Lynx HRP conjugation kit (Bio-Rad) according to manufacturer's protocol. Plates were incubated for an additional hour at room temperature. Subsequently, the plates were washed ten times with PBST, followed by detection with QuantaBlu fluorescence detection reagent (Thermo Fisher).

For the comparison of BiCatcher2 and BiCatcher3, 20 μL of a 1:2 dilution series of the bivalent constructs, starting at a concentration of 2 nM, were transferred onto the plate and allowed to incubate for 1 h at room temperature. Next, the plates were washed five times with PBST before the addition of 20 μL/well of 2 μg/mL anti-Penta-His-HRP (MCA5995P, clone ABD2.2.20, Bio-Rad) detection, prepared in HISPEC assay diluent (Bio-Rad).

## Cell culture

All cell lines used were regularly tested for mycoplasma contamination and tested negative. No authentication of cell lines was performed. No commonly misidentified cell lines were used in this study.

## PD-1/PD-L1 flow cytometry assay

HKB11 PD-1 cells were generated by transfecting HKB11 cells (Bio-Rad) with pMAX vector[41] encoding human PD-1 (UniProt Q15116) and selection with 0.8 mg/mL geneticin (Invivogen).

HKB11 cells (wildtype and PD-1 stable pool) were cultivated in MAC1.0 v110217 medium (Gibco) with 10% heat inactivated FBS (Gibco) and, for the PD-1 overexpressing cells, 0.8 mg/mL geneticin (Invivogen) in 125 mL Optimum Growth flasks (Thomson), shaking at 120 rpm at 37 °C in humidified atmosphere containing 5% $CO_2$.

For the flow cytometry assay, $3 \times 10^4$ cells/well/40 μL were incubated with 5 nM of mono- or bispecific antibody constructs in flow buffer (DPBS w/o Ca and Mg (PAN-Biotech) with 3% heat inactivated FBS (Gibco)) for 1 h at 4 °C in a V-Bottom 384 well plate (Greiner). After washing twice with 40 μL/well flow buffer, the cells were incubated in flow buffer with 1 nM biotinylated human PD-L1 (ACRO Biosystems) for 30 min at 4 °C. Following incubation, cells were washed twice with 40 μL/well flow buffer and stained with streptavidin-PE (Qiagen) for 30 min at 4 °C. Cells were washed twice in 40 μL/well flow buffer and resuspended in 40 μL/well flow buffer with 2.5 μg/mL DAPI (Merck). The median fluorescence intensity of living single cells was determined in triplicates using a ZE5 Cell Analyzer (Bio-Rad) and FCS Express (De Novo Software) for analysis.

## PD-1/PD-L1 immune checkpoint cellular assay

Jurkat-Lucia TCR-hPD-1 effector cells and Raji-APC-hPD-L1 antigen presenting cells were obtained from Invivogen as components of the PD-1/PD-L1 Bio-IC assay (Invivogen, #rajkt-hpd1). The cells were cultivated in IMDM with glutamine and HEPES (Gibco) with 10% heat inactivated FBS (Gibco) in T75 or T175 flasks (Sarstedt) at 37 °C in humidified atmosphere containing 5% $CO_2$. Growth medium was supplemented with 10 μg/mL blasticidin, 250 μg/mL geneticin, 100 μg/mL zeocin and 100 μg/mL hygromycin for Jurkat cells and with 10 μg/mL blasticidin and 250 μg/mL geneticin for Raji cells.

The cellular assay was performed according to vendor instructions. Briefly, 2 days prior to the assay cells were seeded in medium without antibiotics at $5 \times 10^5$ Jurkat cells/mL and at $4 \times 10^5$ Raji cells/mL. On the day of the assay, the cells were pelleted and suspended in medium without antibiotics at $2.2 \times 10^6$ Jurkat cells/mL and at $1.1 \times 10^6$ Raji cells/mL. In 96-well round bottom Nunclon Delta plates (Thermo), per well, 90 μL of each cell suspension was added to 20 μL of antibody solution in PBS and the plates were incubated 6 h at 37 °C in a humidified incubator at 5% $CO_2$. Each antibody concentration was assayed in triplicate on 3 independent plates. Each plate contained cells incubated with PBS without antibody as background control as well as cells treated with 50 nM dostarlimab used for signal normalization among different plates. After 6 h incubation, 20 μL of the suspension of the co-cultured cells was transferred to 96-well white flat bottom TC plate (Greiner) and 50 μL of the working solution of the QUANTI-Luc 4 Lucia/Gaussia reagent (Invivogen) was added. Luminescence was measured immediately afterwards using a Tecan SPARK reader with 100 ms reading time. Luminescent signal measured in each well was expressed as % of signal elicited by 50 nM dostarlimab in the same plate and the mean of results obtained in 3 independent plates was calculated.

## Thermal unfolding

Thermal unfolding of SpyCatcher mutants was measured on a Prometheus nanoDSF device (Nanotemper) from 20 °C to 90 °C with a ramp rate of 2 °C/minute.

## Computational modeling

Modeling of mutated SpyCatchers and SpyLocks was performed using AlphaFold 2 and OpenMM[42,43]. First, all mutations from SpyCatcher003 in the ordered region of SpyCatcher001 (residues 22–103) were grafted into the SpyCatcher001 structure 4MLI using AlphaFold2[20]. The resulting base model was then mutated to introduce each of eight possible biotinylated cysteine mutants I27C, F29C, A42C, M44C, S59C, F75C, Y84C, and F92C using PDBFixer[43]. To model biotinylated cysteine, a residue was initially parameterized with the General Amber Force Field (GAFF) via OpenMM's residue template generator, then modified to incorporate partial charges from the disulfide-bonded cysteine CYX residue and protein parameters from Amber ff14SB[44]. The resulting mutant models were solvated in a bounding box with a minimum of 2 nm padding distance, and Na$^+$ and Cl$^-$ ions were added to neutralize the system up to an ionic strength of 0.15 M. Each system was energy minimized and equilibrated with OpenMM at 300 K for 2 ps with 2 fs steps, followed by simulation in an isothermal-isobaric (NpT) ensemble for an additional 10 ns. For simulations of the SpyTag-bound SpyCatcher003 complex, the S59C mutant, and its biotinylated version, the SpyTag peptide coordinates were initially chosen from 4MLI. Tag-bound models were generated for each condition, solvated, and energy minimized as above, except the temperature was gradually raised from 300 to 350 K during equilibration and then simulated for 100 ns at 350 K. Five replicate simulations were performed for each condition. In all simulations, the cut-off for nonbonded interactions was set at 1.2 nm, and the Particle Mesh Ewald (PME) method was used for electrostatic long-range interactions. Visualizations were generated using VMD or UCSF Chimera[45,46].

## Statistics & reproducibility

We did not perform any statistical tests. No statistical method was used to predetermine sample size. No data were excluded from the analyses. The experiments were not randomized. The investigators were not blinded to allocation during experiments and outcome assessment.

## Reporting summary

Further information on research design is available in the Nature Portfolio Reporting Summary linked to this article.

## Data availability

The authors declare that the data supporting the findings of this study are available within the paper and the supplementary information. The previously published structure 4MLI can be found here: 4MLI. Uniprot entry Q15116 can be found here: Q15116. Source data are provided with this paper.

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

## Acknowledgements
We thank our colleagues in the protein production and QC teams for generating and validating the reagents used in this study. We thank Achim Knappik for critically reading the manuscript. This work was funded internally by Bio-Rad Laboratories, Inc, with no external funding sources.

## Author contributions
Conceptualization and design of experiments: C.H., M.P., F.Y. Performed experiments: H.H., M.P., S.J.K., M.W., M.C. Generated reagents: W.P., S.H. Molecular modeling: V.S.L. Writing: C.H., M.P., F.Y. with input from all authors.

## Competing interests
M.P., C.H., and F.Y. are inventors on a patent application (US Patent and Trademark Office application number 18/655,747) pertaining to the SpyLock technology described here. All authors except VSL are employees of Bio-Rad AbD Serotec GmbH. The remaining authors declare no competing interests.
