## [Peer Review File · Nature Communications]

REVIEWER COMMENTS

Reviewer #1 (Remarks to the Author):

Hentrich et al. demonstrate a relatively simple and powerful approach to gain redox control over SpyCatcher/SpyTag protein-protein ligation. Following a cysteine scan at each position within SpyCatcher003's structured regions, several mutants were identified that retained high ligation activity with SpyTagged proteins. Of these, 5 mutants were identified that exhibited dramatically decreased activity following S-thiolation (either with TNB or HPDP-biotin). Functional activation of these "SpyLock" species was obtained with small molecule reductant (TCEP) treatment. Building on this success, the authors then fuse SpyLock with SpyCatcher, creating a species termed "BiLock", allowing for sequential coupling of distinct SpyTagged species. This method was used to create bispecific antibodies that are demonstrated as active with a luciferase-based cell reporter assay. Overall, I found the studies demonstrating SpyLock's activation and BiLock's utility well described and properly controlled. An appropriate amount of detail appears to have been included in the methods section, and the supporting information provides useful information that complements the main text figures. Further, the authors do an excellent job highlighting the potential benefits of their strategy (notably describing their process design's applicability to redox-control other proteins) as well as their weaknesses as part of their extended discussion. I very much enjoyed reading this paper and expect that it will be of interest to the Nature Communications' readership. I recommend the following changes be made to the manuscript prior to potential acceptance:

- In general, the quality of main text figures could be improved significantly. The underlying methods and obtained data are quite exciting; I would hope that the figure aesthetics could match this. I would further encourage the authors to update the figure graphics such that their main takeaways can be mostly gathered without having to read the captions.
- Figure 1c could be expanded to show cysteine regeneration upon TCEP treatment as this is central to SpyLock's activation.
- The authors do a good job of explaining how their methods may not have good compatibility with disulfide-containing proteins, given the reducing agent. It would be helpful for the authors to also comment on any limitations imposed onto any potential SpyLock fusion partners that would also be subjected to potential S-thiolation.
- SpyLock rate constants should be determined explicitly (both in closed and open confirmation) and compared with those of the various SpyCatcher variants. This will better support general statements about SpyLock's reduced activity relative to SpyCatcher that are made throughout the text.
- It was not clear why SpyTag002 was employed in conjunction with SpyTag003, the latter of which was evolved to work with the used SpyCatcher003. The authors should justify this choice in the main text.

- These proteins were named “SpyCatcher003” (vs “SpyCatcher3”) and “SpyTag002” (vs “SpyTag2”), etc, in their original reports. The authors should consider updating their paper to be more consistent with the names elsewhere in the literature.
- Purified expression yields of final species, and assembly yields in the case of BiLock bispecific antibody generation, should be given.

Reviewer #2 (Remarks to the Author):

The manuscript by Hentrich et al. (NCOMMS-23-51874) describes a technical improvement on the previously described “Spytag/SpyCatcher” technology to create the “SpyLock” platform, by engineering functional control into the technology to enable the rapid and high-throughput ligation of independent proteins. Authors then exemplify the potential of this technological improvements by combining the SpyLock and SpyCatcher platforms to rapidly generate bi-specific PD-1/PD-L1 bispecific antibodies that demonstrate in-vitro functional activity.

While the described technical advances and are certainly of interest, it is hard to understand how generally applicable and valuable this platform will be.

Reviewer #3 (Remarks to the Author):

The authors describe a method to allosterically and reversibly inhibit the reaction between SpyCatcher and SpyTag through the insertion of cysteine mutants modified with either DTNB or HPDP Biotin. They demonstrate the use of this technology to generate bispecific antibodies.

Regarding significance - the ability to regulate protein function through the engineering of allosteric sites is significant with potentially broad applicability if translatable across protein classes. Here the authors describe a specific example of this (SpyCatcher) and potentially need more data to demonstrate some of their key claims (in detail below). Additionally, while regulation of SpyCatcher function could have many applications, as the authors note, this is not the first disclosure of such a technology (ref #20, #21).

Specific claims that need additional exposition:

1) The inhibition is allosteric (across the 8 mutants that were prioritized):

-Several of the cysteine mutations evaluated are in the central hydrophobic binding groove of the SpyCatcher protein. This groove is where SpyTag binds to SpyCatcher. Would it truly be called allosteric inhibition if one were to disrupt the central binding groove and the binding site of SpyTag through modification?

-The authors show that there is still a measured T_m after modification of mutated residues. However, the change in T_m is quite significant for many of these mutations, to the point that it might not be accurate to call this inhibition allosteric as much as it is reversible (or irreversible) inhibition of proper folding.

-The mechanisms of inhibition for all 8 sites prioritized may be quite distinct. This is evident from Figure 2B, and the relative measured melting temperatures pre and post modification of engineered cysteines with HPDP biotin, calling this allostery is perhaps an over-simplification.

-If the presence of charge is irrelevant to the inhibition, one might argue that this is steric inhibition of SpyTag binding to SpyCatcher as opposed to allostery, for many of the mutations tested.

-For the one mutation the authors focus on (S59C) the positioning is removed enough from the main groove that allostery seems like the most likely mechanism - could the authors please model S59C with HPDP biotin or DTNB bound (for supplementary figure 10)?

2)The inhibition is reversible and the integrity of the protein is not affected by reduction:

-The authors show melting temperatures of the unmodified and modified/reacted mutants in figure 2B. They additionally show supporting SDS-PAGE gels indicating reactivity post reduction. However, the authors exclude any measurements of melting temperature post reduction of the HPDP-biotin. Is the protein still equivalently stable in the presence of TCEP/post reduction? This data is needed to demonstrate the integrity of the protein despite reaction conditions.

-The authors discuss the use of TCEP in the presence of Fabs in order to generate bispecific antibodies. A few points:

The authors treat Fab fragments with TCEP and evaluate binding via ELISA (Sup Fig 7) and claim minor effects on antigen binding, however, some of the observed differences appear major (for example, AbD51340ad).

-The authors quench the reduction with Bis-PEG3-azide, however, would it not be more effective to purify the final product instead? Additionally, could the authors treat SpyLock with TCEP, react with SpyTag, purify, then reoxidize (and purify again) to generate fully folded Bispecific Fabs (with properly oxidized disulfide bonds)? The current route is at risk to generate bispecific antibodies with reduced binding propensity and reduced stability versus other routes.

-Could the authors verify that Fabs still bind to their targets with equal affinity by SPR?

Hentrich *et al*, Engineered Reversible Inhibition of SpyCatcher Reactivity Enables Rapid Generation of Bispecific Antibodies

Response to Reviewers

Reviewer #1 (Remarks to the Author):

Hentrich et al. demonstrate a relatively simple and powerful approach to gain redox control over SpyCatcher/SpyTag protein-protein ligation. Following a cysteine scan at each position within SpyCatcher003's structured regions, several mutants were identified that retained high ligation activity with SpyTagged proteins. Of these, 5 mutants were identified that exhibited dramatically decreased activity following S-thiolation (either with TNB or HPDP-biotin). Functional activation of these "SpyLock" species was obtained with small molecule reductant (TCEP) treatment. Building on this success, the authors then fuse SpyLock with SpyCatcher, creating a species termed "BiLock", allowing for sequential coupling of distinct SpyTagged species. This method was used to create bispecific antibodies that are demonstrated as active with a luciferase-based cell reporter assay. Overall, I found the studies demonstrating SpyLock's activation and BiLock's utility well described and properly controlled. An appropriate amount of detail appears to have been included in the methods section, and the supporting information provides useful information that complements the main text figures. Further, the authors do an excellent job highlighting the potential benefits of their strategy (notably describing their process design's applicability to redox-control other proteins) as well as their weaknesses as part of their extended discussion. I very much enjoyed reading this paper and expect that it will be of interest to the Nature Communications' readership.

We thank the reviewer for the positive evaluation of our work.

I recommend the following changes be made to the manuscript prior to potential acceptance:

- In general, the quality of main text figures could be improved significantly. The underlying methods and obtained data are quite exciting; I would hope that the figure aesthetics could match this. I would further encourage the authors to update the figure graphics such that their main takeaways can be mostly gathered without having to read the captions.

We have worked on making the figures more immediately accessible and also tried to improve the aesthetics.

- Figure 1c could be expanded to show cysteine regeneration upon TCEP treatment as this is central to SpyLock's activation.

We have done so.

- The authors do a good job of explaining how their methods may not have good compatibility with disulfide-containing proteins, given the reducing agent. It would be helpful for the authors to also comment on any limitations imposed onto any potential SpyLock fusion partners that would also be subjected to potential S-thiolation.

We have added a sentence to the discussion, explaining this potential limitation.

- SpyLock rate constants should be determined explicitly (both in closed and open confirmation) and compared with those of the various SpyCatcher variants. This will better support general statements about SpyLock's reduced activity relative to SpyCatcher that are made throughout the text.

We have measured the rate constant for all SpyLocks and SpyCatcher001/3 and present these data in supplementary figure 5 and supplementary table 1.

- It was not clear why SpyTag002 was employed in conjunction with SpyTag003, the latter of which was evolved to work with the used SpyCatcher003. The authors should justify this choice in the main text.

We have observed that antibody yield of antibodies equipped with SpyTag002 is slightly higher than with SpyTag003, and therefore by default produce our antibodies with SpyTag002, especially since reaction speed is orders of magnitude higher than required for antibody conjugation at micromolar concentrations and above. We still profit from using SpyCatcher003s, as it was shown that SpyTag002 reacts faster with SpyCatcher003 than SpyCatcher002 (Keeble AH, et al. Approaching infinite affinity through engineering of peptide-protein interaction. Proc Natl Acad Sci USA, 2019). We now mention this briefly in the text.

- These proteins were named "SpyCatcher003" (vs "SpyCatcher3") and "SpyTag002" (vs "SpyTag2"), etc, in their original reports. The authors should consider updating their paper to be more consistent with the names elsewhere in the literature.

We have updated the nomenclature to the double-O names.

- Purified expression yields of final species, and assembly yields in the case of BiLock bispecific antibody generation, should be given.

The yield of the rapid conjugation is close to 100% with regard to the limiting reagent, as seen in figure 5. We have performed new experiments with sfGFP-SpyTag003 to measure

the yield of the purification steps for the slow assembly protocol (suppl. figure 11). We have also added the yield of BiLockCatcher expression to the methods section.

Reviewer #2 (Remarks to the Author):

The manuscript by Hentrich et al. (NCOMMS-23-51874) describes a technical improvement on the previously described "Spytag/SpyCatcher" technology to create the "SpyLock" platform, by engineering functional control into the technology to enable the rapid and high-throughput ligation of independent proteins. Authors then exemplify the potential of this technological improvements by combining the SpyLock and SpyCatcher platforms to rapidly generate bi-specific PD-1/PD-L1 bispecific antibodies that demonstrate in-vitro functional activity.

While the described technical advances and are certainly of interest, it is hard to understand how generally applicable and valuable this platform will be.

We thank the reviewer for finding our work interesting.

Reviewer #3 (Remarks to the Author):

The authors describe a method to allosterically and reversibly inhibit the reaction between SpyCatcher and SpyTag through the insertion of cysteine mutants modified with either DTNB or HPDP Biotin. They demonstrate the use of this technology to generate bispecific antibodies.

Regarding significance - the ability to regulate protein function through the engineering of allosteric sites is significant with potentially broad applicability if translatable across protein classes. Here the authors describe a specific example of this (SpyCatcher) and potentially need more data to demonstrate some of their key claims (in detail below). Additionally, while regulation of SpyCatcher function could have many applications, as the authors note, this is not the first disclosure of such a technology (ref #20, #21).

We thank the reviewer for recognizing the significance of our work. We have done extensive additional work, including a new collaboration with an expert in molecular modeling, to address the concerns of the reviewer. Regarding novelty, we have highlighted in the text that our technology is much more practical than the previous disclosures. We would also like to point out that it could be argued that directly modifying the functional residue of an enzyme with a removable protection group is not a particularly innovative conceptual advance, whereas modifying further removed amino acids to regulate protein function, while also not entirely without precedent, opens up a plethora of interesting scientific questions, some of which we have tried to address.

Specific claims that need additional exposition:

1) The inhibition is allosteric (across the 8 mutants that were prioritized):

-Several of the cysteine mutations evaluated are in the central hydrophobic binding groove of the SpyCatcher protein. This groove is where SpyTag binds to SpyCatcher. Would it truly be called allosteric inhibition if one were to disrupt the central binding groove and the binding site of SpyTag through modification?

Just as the reviewer, we also find the question of allosteric regulation and the applicability of this concept to SpyLock extremely interesting. That is why we originally discussed some of this background in the introduction. We have shortened this substantially to reduce the overall word count.

Furthermore, we agree that just based on the original data, it is very difficult to deduce whether the regulation is indeed allosteric in the commonly understood sense. That is why we had only speculated about this possibility in one sentence in the discussion. However, the new modeling work that was inspired by this reviewer does indeed show the possibility for allosteric regulation for at least some SpyLocks, and we discuss this possibility.

-The authors show that there is still a measured T_m after modification of mutated residues. However, the change in T_m is quite significant for many of these mutations, to the point that it might not be accurate to call this inhibition allosteric as much as it is reversible (or irreversible) inhibition of proper folding.

It is indeed an interesting question from a science philosophy point of view, whether steric obstruction away from the reactive site could be called allosteric or not. Likewise, whether the reversible induction of partial unfolding could be considered allosteric regulation is unclear. At least for us, it is impossible to answer these questions definitely, as there is to our knowledge no commonly agreed upon definition of allostery and the meaning of the term has substantially changed over the decades.

Therefore, in the revised version, we only speculate about allostery for the 2 SpyLocks where the modeling showed no obstruction of the SpyTag binding pocket. We do not think these modified mutants are fully unfolded, as we can still measure a T_m for them. Instead, the structural changes we observe in simulations (see below) might be responsible for the observed destabilization of these two mutants.

-The mechanisms of inhibition for all 8 sites prioritized may be quite distinct. This is evident from Figure 2B, and the relative measured melting temperatures pre and post modification of engineered cysteines with HPDP biotin, calling this allostery is perhaps an over-simplification.

Based on our modelling work, we can now hypothesize which SpyLocks act in which way, and do so in the text.

-If the presence of charge is irrelevant to the inhibition, one might argue that this is steric inhibition of SpyTag binding to SpyCatcher as opposed to allostery, for many of the mutations tested.

We could unfortunately not model the inhibition with the charged modification through DTNB (see below). But for the A42C and S59C mutants, our simulations suggest an aspect of allosteric regulation even in the absence of a charge.

-For the one mutation the authors focus on (S59C) the positioning is removed enough from the main groove that allostery seems like the most likely mechanism - could the authors please model S59C with HPDP biotin or DTNB bound (for supplementary figure 10)?

We could not model the DTNB due to this modification being very difficult to parametrize within our molecular dynamics framework because of its partial charges. We did however manage to implement modeling of the larger, but uncharged HPDP-biotin derived modification (methods). We reproducibly saw the alkane chain fold outwards through a solvent accessible pore and not occupying the binding groove. This might explain why the drop in T_m was most pronounced for this mutant, with the biotin disturbing the fold. In this case it is therefore likely that we are indeed observing allostery as applied to monomeric proteins changing their conformation subtly after posttranslational modification.

We have also modeled the other SpyLocks, present these simulations in figure 4, and discuss the results in the text. Interestingly, the other SpyLock for which we propose allostery as a possible mechanism has the second highest drop in T_m after modification with HPDP-biotin.

2)The inhibition is reversible and the integrity of the protein is not affected by reduction:

-The authors show melting temperatures of the unmodified and modified/reacted mutants in figure 2B. They additionally show supporting SDS-PAGE gels indicating reactivity post reduction. However, the authors exclude any measurements of melting temperature post reduction of the HPDP-biotin. Is the protein still equivalently stable in the presence of TCEP/post reduction? This data is needed to demonstrate the integrity of the protein despite reaction conditions.

We performed these measurements and included them in figure 2B. The T_m indeed goes back to its original value post-reduction.

-The authors discuss the use of TCEP in the presence of Fabs in order to generate bispecific antibodies. A few points:

The authors treat Fab fragments with TCEP and evaluate binding via ELISA (Sup Fig 7) and

claim minor effects on antigen binding, however, some of the observed differences appear major (for example, AbD51340ad).

We have updated the discussion of the ELISA to reflect this more accurately. However we would like to point out that these changes occurred after a full week of incubation with TCEP, a scenario that does not occur with our methods. When incubating with TCEP overnight, the maximum period that would occur for convenience when producing a larger number of bispecific antibodies, we do not see a measurable effect of TCEP (see below).

-The authors quench the reduction with Bis-PEG3-azide, however, would it not be more effective to purify the final product instead? Additionally, could the authors treat SpyLock with TCEP, react with SpyTag, purify, then reoxidize (and purify again) to generate fully folded Bispecific Fabs (with properly oxidized disulfide bonds)? The current route is at risk to generate bispecific antibodies with reduced binding propensity and reduced stability versus other routes.

This would certainly be possible, but the purpose of our approach was to have a rapid protocol. Given that we cannot detect any effect on affinity (below) of the low concentrations of TCEP used over the time period of a typical coupling, we do not think reoxidation is necessary, and therefore did not perform this experiment. If the azide or other quenching reagents are not desirable in the final product (even though we did not see any effect in our cell assay), one could certainly perform a final buffer exchange. We now mention this possibility in the text.

-Could the authors verify that Fabs still bind to their targets with equal affinity by SPR?

This is indeed an important control experiment, and we thank the reviewer for this suggestion. We incubated three different antibodies overnight in absence or presence of the TCEP concentration used in our protocol. The affinities as measured by BLI are indeed identical. We present this data in supplemental figure 8.

REVIEWERS' COMMENTS

Reviewer #1 (Remarks to the Author):

The authors have satisfactorily addressed my prior concerns. I am pleased to recommend publication of this manuscript in Nature Communications at this time.

Reviewer #3 (Remarks to the Author):

This reviewer thanks the authors for submitting thoughtful responses to all comments and agrees that the MD simulations clarify the MOA questions. The clarifying edits were much appreciated and no further revisions are requested.